# Effects of the Use of Good Agricultural Practices on Aflatoxin Levels in Maize Grown in Nandi County, Kenya

**Grace Nkirote Marete** [1,*], **Laetitia Wakonyu Kanja** [1] , **James Mucunu Mbaria** [1],
**Mitchel Otieno Okumu** [1] , **Penina Afwande Ateku** [1], **Hannu Korhonen** [2] **and Vesa Joutsjoki** [2]

[1]  Department of Public Health, Pharmacology, and Toxicology, Faculty of Veterinary Medicine,
    University of Nairobi, P.O. Box 29053-00625, Nairobi 00100, Kenya; lkanja@uonbi.ac.ke (L.W.K.);
    james.mbaria@uonbi.ac.ke (J.M.M.); mytchan88@gmail.com (M.O.O.); pateku@uonbi.ac.ke (P.A.A.)
[2]  National Resources Institute, P.O. Box 2 FI-00791, 00791 Helsinki, Finland; hannu.j.korhonen@luke.fi (H.K.);
    vesa.joutsjoki@luke.fi (V.J.)
*  Correspondence: gracemarete@yahoo.com; Tel.: +254-718732869

**Abstract:** Aflatoxin contaminated maize is of public health concern in Kenya. Training farmers on good agricultural practice (GAP) has been touted as a mitigative measure. Little is known of the effect of such training on aflatoxin levels in maize grown in Kenya. This study evaluated what effect training farmers on GAP has on aflatoxin levels in maize grown in Kaptumo, Kilibwoni, and Kipkaren divisions in Nandi County. Ninety farmers were recruited for the study and interviewed on GAP. Maize samples were additionally collected from the participating farmers and analyzed for aflatoxins using competitive enzyme-linked immunosorbent assay (c-ELISA). All farmers prepared the land before planting, applied correct spacing between the planted crops, carried out weeding, cleaned their stores before use, checked the condition of the maize after harvesting, sorted maize after shelling, and knew about aflatoxins. The majority of the farmers (90%) used fertilizers, dried maize after harvesting, knew that aflatoxins were harmful to humans, and used clean transport in transporting the harvested maize. About 98% of farmers did stooking after harvesting and 97% used wooden pallets in the maize stores. The percentage of farmers who practiced early planting, top dressing, crop rotation, raising stores above the ground, applying insecticide after shelling and feeding damaged/rotten seeds to their animals was 84–96%, 62–80%, 67–85%, 86–98%, 63–81%, and 7–21% respectively. About 18/90 (20%) of all farmers reported that they had a relative who had died from liver cancer, and the mean aflatoxin levels in season 1 were significantly different from those in season 2 (1.92 ± 1.07 ppb; 1.30 ± 1.50 ppb). Our findings suggest that although training farmers to adopt good agricultural practices was observed to be efficient in mitigating the problem of aflatoxins, the receptiveness of farmers to different aspects of the training may have differed. Therefore, in designing an optimized regional aflatoxin contamination strategy, local applicability should be considered.

**Keywords:** good agricultural practice; aflatoxin; Nandi County; mycotoxins; Kenya; maize

## 1. Introduction

Aflatoxins are a group of mycotoxins that are produced by soil-borne fungi (*Aspergillus* species) that are ubiquitous in nature [1]. The major types are $AFB_1$, $AFB_2$, $AFG_1$, and $AFG_2$ [1]. Most species of *Aspergillus* produce B-type aflatoxins, although species related to *A. parasiticus* and *A. nomius* can produce G-type aflatoxins [1]. The fungus may be recognized by a grey-green or yellow-green mold growing on corn kernels in the field or storage [2]. Aflatoxins are carcinogenic, mutagenic,



and immunosuppressive in both humans and animals [1]. At high levels of exposure, they can cause acute toxicity and potentially death in mammals, birds, and fish [1].

Aflatoxins have received quite a bit of attention in Kenya over the last three decades [3–9]; the most recent was the case of aflatoxin-contaminated peanut butter [10–12]. Associations between the nutritional status of children and dietary exposure to aflatoxins have been reported in studies conducted in Nairobi and Kisumu [13,14]. Children under 30 months of age in one county (Makueni) have been reported to have 1.4 times higher levels of aflatoxin $M_1$ in urine than those of the same age in another county (Nandi) [7]. It has also been reported that the levels of aflatoxin and fumonisins (a group of mycotoxins derived from Fusarium) in animal feed, milk, and sorghum in selected areas of Kenya have exceeded the critical limits set by the Kenya Bureau of Standards and WHO/FAO [5,7,15].

Several toxigenic strains of *Aspergillus* species have been isolated from maize kernels from two maize growing areas of the country (Nandi and Makueni) [3]. Moreover, a study on Kenyan dairy farmers' perception of molds and mycotoxins reported that farmers had a low understanding of the dangers of mycotoxins in food, and in some instances, farmers carried out practices that increased the risk of exposure to mycotoxins [16]. At the same time, groups that were aware of aflatoxin were not always engaging in risk mitigation [16].

Death resulting from acute aflatoxicosis has also been reported in the country [17–19]. In 1981, there were 20 cases of acute aflatoxicosis in parts of the former Eastern province [17]. This was occasioned by a severe drought followed by heavy rains during the harvesting period [17]. Between January and June 2004, 125 of 317 people died in Makueni and Kitui districts in Kenya after presenting to various hospitals with symptoms of acute aflatoxicosis [18]. Investigations revealed that the concentration of aflatoxin $B_1$ in some maize samples from the area was a staggering 220 times above the 20 ppb critical limit for food set by the Kenya Bureau of Standards [18]. Following this outbreak, a 2004 study set out to determine the extent of regional contamination and the status of maize in commercial markets [19]. This study reported that 50% of maize products sampled had aflatoxin levels greater than the Kenyan regulatory limit (20 ppb at the time). The same study also reported that 35% of maize samples tested had levels >100 ppb and 7% had levels >1000 ppb [19].

Barely a year after the 2004 outbreak, 32 out of 75 people with symptoms of acute aflatoxicosis in the same area (Makueni and Kitui) died following aflatoxin exposure [20]. Investigations revealed that locally produced maize from subsistence farming was the likely source of contamination that resulted in the two outbreaks [20].

In 2010, two and a half million bags (each 90 kg in weight) of aflatoxin-contaminated maize was seized by authorities in Kenya [1]. A year later, Proctor and Allan East Africa (a major cereals manufacturer in the region) recalled 25 tonnes of relief food (Unimix; a high-protein maize meal) after it was discovered to be contaminated with aflatoxins [21]. The contaminated batch was to be sent to areas in the country adversely hit by drought [21].

While all these published works provide valuable information and recommendations, there has been very little research in the form of measures aimed at mitigating this public health problem. Sound agronomic and postharvest practices including training farmers on good agricultural practice may reduce aflatoxin contamination in maize. However, little is known of the effect of such training on aflatoxin levels in maize grown in Kenya. The present study aimed to determine what effect training farmers in Nandi County on good agricultural practice may have had on aflatoxin levels in maize grown in the region.

## 2. Materials and Methods

### 2.1. Study Area

Nandi County is part of the former Rift Valley Province in Kenya, and covers an area of about 2900 km$^2$ [3,22]. It lies between longitude 35°08′60.00″ E and latitude 0°10′0.00″ N [23] and is elevated between 1000 and 2000 m above sea level [3]. The area receives about 1000–1500 mm of rainfall on

average, with a mean temperature of about 20 °C [3]. Kaptumo, Kilibwoni, and Kipkaren divisions in the county (Figure 1) were selected as study areas as these have been reported to be the areas where maize cultivation is actively practiced [3].

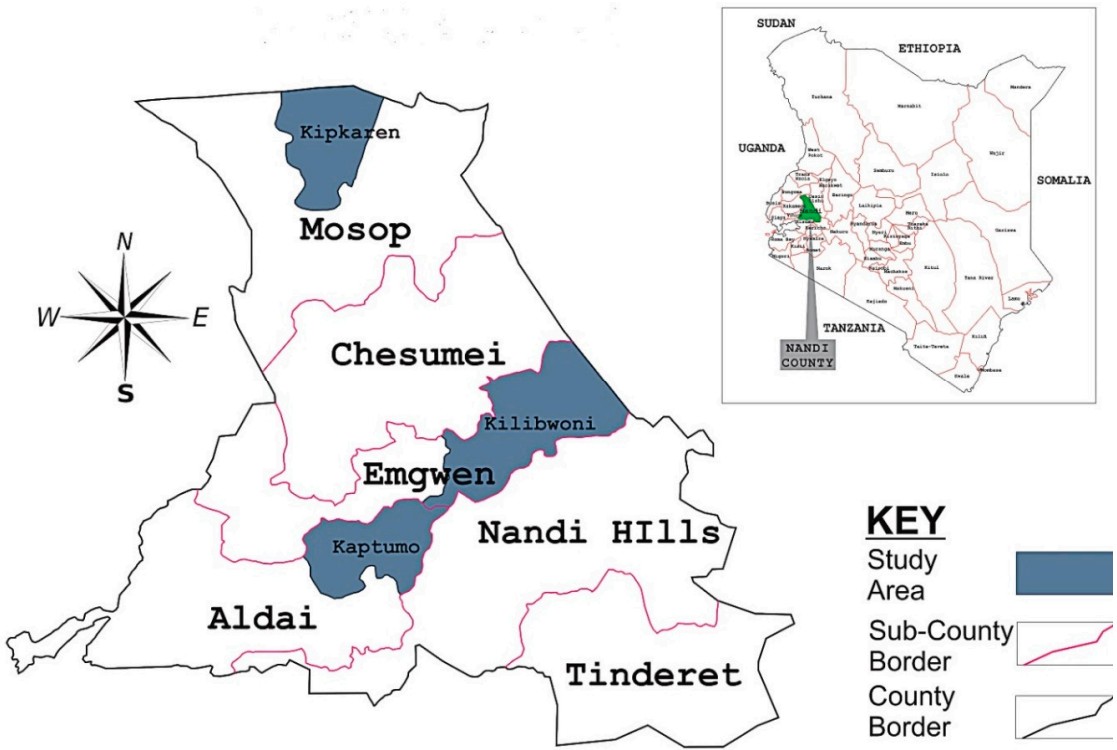

**Figure 1.** Map of Nandi County showing the study areas.

### 2.2. Study Design

This was a cross-sectional study carried out in two maize planting seasons in Nandi County (January–November 2016; season 1, and January–November 2017; season 2). The study involved a cross-sectional survey of farmers' knowledge of good agricultural practice in selected divisions of Nandi County.

Farmers who were willing to participate in the survey and were above 18 years of age were considered eligible for the study. All participants in selected FFSs were given the chance to complete the questionnaire and were divided into six farmer field schools: Mwangaza and Keteba in Kaptumo division, Sobetab Gaa and Toret Gaa in Kipkaren division, and Toleltany and Kisob Katanin in Kilibwoni division. A questionnaire was then administered to all the recruited farmers, and for each farmer interviewed, a maize sample was also collected for aflatoxin analysis at the Mycotoxin Research Centre at the Department of Public Health, Pharmacology and Toxicology, University of Nairobi.

### 2.3. Data Collection

A 33-item questionnaire was designed for use in this study (Appendix A). Each question was read to each farmer to ensure that they understood what was being asked. The questionnaire consisted of the following sections.

#### 2.3.1. Section 1: Socio-Demographic Information

Seven questions were contained in this section and inquired about general information, including the gender of the farmers, his/her division, the name of the farmer field school, and the

number of years the farmer had practiced farming. Other questions included the acreage of the farms, whether the farmers had other sources of income, and the nature of the farming practiced.

### 2.3.2. Section 2: Questions Related to Good Agricultural Practice

This section contained 26 questions which included inquiries on top dressing, stooking, whether the stores were raised above the ground and had wooden pallets, and the level of awareness farmers had concerning aflatoxins. Other questions touched on plant spacing, the use of fertilizer and manure, quantities of fertilizer used per acre, how dry the maize was at harvesting time, and how the maize was harvested. There were also questions on whether harvested maize was put on mats/racks or the ground, whether the maize eventually went to the store or the ground of the house and whether the farmers checked on the maize in the store to see whether it was still in good condition. Moreover, questions on early planting, land preparation, weeding and cleaning of the stores were asked. Other questions included the time the maize was dried, sorting practices and when this was done, the application of insecticides, crop rotation, and whether they were feeding maize seeds that were damaged to animals or not. Finally, the means of transport used by the farmers was also determined.

### 2.4. Sample Collection

About 2 kg of shelled maize kernels were collected from each interviewed farmer by use of a closed spear driven through the upper parts and sides of bags containing maize kernels. This was then transferred to sterile bags which were then sealed, and each bag was assigned a unique identification code. The sealed bags were stored away from heat and sunlight and transported to the Department of Public Health, Pharmacology and Toxicology of the University of Nairobi, awaiting analysis.

### 2.5. Preparation of Maize Samples and Analysis of the Levels of Aflatoxin

The samples were ground by use of a grinder (Grindomix® G200; Retsch GmbH 42781, Haan, Germany) to a fine powder at the mycotoxin research center of the University of Nairobi. Two grams of the ground samples were accurately weighed by the use of a weighing balance (Shimadzu Corp TX 42021) and transferred to 50 mL falcon tubes (Thermo Fisher Scientific, Waltham, MA, USA). Aflatoxin extraction was performed by adding 70% methanol to the 2 g of maize powder and the falcon tubes were manually shaken for 10 min. The falcon tubes were then centrifuged for 5 min at 3000 revolutions per min. One hundred microliters of the supernatant were collected and transferred to a 2 mL Eppendorf tube and 600 µL of distilled water was added to the supernatant and the mixture vortexed by use of a Wisemix® VM10-Korea vortex machine.

### 2.6. Determination of Aflatoxin Levels in the Prepared Samples

The total aflatoxin levels in the prepared samples and standard dilutions were tested using a competitive enzyme-linked immunosorbent assay (ELISA) kit (Ridascreen® Aflatoxin Total, R4701, R-Biopharm AG, Darmstadt, Germany), as per manufacturer's instructions [24].

Briefly, 50 µL of the prepared sample extracts and standard dilutions were pipetted in duplicate into 96-well microtiter plates. Fifty microliters of the conjugate were then added to each well, followed by 50 µL of the antibody. The contents of the plate were then mixed by gently shaking the plate manually. The resultant mixture was then incubated for 30 min at room temperature. The wells were then emptied by tapping the microwell holder upside down vigorously against an absorbent paper (three times in a row) to ensure complete removal of liquid from the wells. The wells were then filled with 250 µL phosphate buffer solution. The wells were again emptied, and this was repeated two more times. One hundred microliters of the substrate were then added to each well and the plate was shaken gently to mix the contents. The plate was then incubated for 15 min at room temperature. One hundred microliters of stop solution was then added to each well and absorbance measurements recorded at 450 nm on a multiplate reader (Thermo Electron Corporation Multiskan EX, Waltham, MA, USA).

Total aflatoxins levels were calculated using ELISA software (Rida® Soft, Z9999, R-Biopharm AG, Darmstadt, Germany).

## 2.7. Data Analysis

Aflatoxin levels in maize samples were expressed as the mean ± standard deviation, and the student's t-test was used for comparison between season 1 and 2 (Appendix B). $p < 0.05$ was considered significant. Moreover, the adjusted Wald technique [25] was used to determine the confidence interval for responses provided by farmers in Nandi County.

## 3. Results

### 3.1. Socio-Demographics of Maize Farmers Interviewed in the Study Area

Table 1 is a summary of the socio-demographic profile of farmers in the study area.

**Table 1.** Sociodemographic characteristics of farmers interviewed in the study area.

| Variable | Frequency | % |
|---|---|---|
| **Gender** | | |
| Male | 38 | 42.2 |
| Female | 52 | 57.8 |
| **Number of Years in Farming** | | |
| 1–5 years | 74 | 82.2 |
| 6–10 years | 4 | 4.4 |
| 11–15 years | 3 | 3.3 |
| >15 years | 9 | 10.0 |
| **Months When Maize was Planted** | | |
| January | 22 | 24.4 |
| March | 61 | 67.7 |
| April | 7 | 7.8 |
| **Acreage of Farm** | | |
| <1 acre | 11 | 12.2 |
| 1–5 acres | 65 | 72.2 |
| 5.5–20 acres | 14 | 15.6 |
| **An alternative Source of Income** | | |
| Yes | 54 | 60.0 |
| No | 36 | 40.0 |
| **Reasons for Farming** | | |
| Subsistence | 24 | 26.7 |
| Commercial | 66 | 73.3 |

Ninety farmers were interviewed and were drawn from the Kaptumo (23, 25.6%), Kilibwoni (34, 37.8%) and Kipkaren (33, 36.7%) divisions. The distribution of the 90 farmers among the farmer field schools was as follows: 12 (13.3%) from Keteba, 11 (12.2%) from Mwangaza, 18 (20%) from Kisob Katanin, 16 from Toleltany (17.8%), 18 (20%) from Sobetab Gaa, and 15 (16.7%) from Toret Gaa.

### 3.2. Overview of Good Agricultural Practices by Farmers in the Study Area

Table 2 is a summary of the agricultural measures adopted by farmers in the study area. Crop rotation had the highest confidence intervals, i.e., 0.86 to 0.98, while the practice of feeding damaged/rotten seeds to animals had the lowest confidence intervals, i.e., 0.07 to 0.021.

**Table 2.** Agricultural measures adopted by farmers in Nandi County and their confidence intervals.

| Measure | Frequency (n = 90) | Proportion | Margin of Error | 95% Confidence Interval | Percentage (%) |
|---|---|---|---|---|---|
| Early planting | 81 | 0.90 | 0.06 | 0.84–0.96 | 84–96% |
| Top dressing | 64 | 0.71 | 0.09 | 0.62–0.80 | 62–80% |
| Raising stores above the ground | 68 | 0.76 | 0.09 | 0.67–0.85 | 67–85% |
| Crop rotation | 83 | 0.92 | 0.06 | 0.86–0.98 | 86–98% |
| Applying insecticide after shelling | 65 | 0.72 | 0.09 | 0.63–0.81 | 63–81% |
| Feeding damaged/ rotten seeds to animals | 13 | 0.14 | 0.07 | 0.07–0.21 | 7–21% |
| Harvesting during rain | 43 | 0.48 | 0.10 | 0.38–0.58 | 38–58% |

### 3.3. Overview of other Variables Captured in the Questionnaire

Table 3 is a summary of the distribution of other variables captured in the questionnaire. Small-sized stooks were the most commonly used stooks, most farmers described maize seeds to be moderately dry, and most farmers used stores. Table 3.

**Table 3.** Overview of other variables captured in the questionnaire.

| Variable | Frequency (n = 90) | % |
|---|---|---|
| **Nature of Maize Seeds** | | |
| Very dry | 11 | 12.2 |
| Moderately dry | 79 | 87.8 |
| **Size of the Stook** | | |
| Small | 85 | 94.4 |
| Big | 3 | 3.3 |
| Not done | 2 | 2.2 |
| **Storage after Harvesting** | | |
| Mat/rack | 45 | 50.0 |
| On the ground | 45 | 50.0 |
| **Storage Facility Used** | | |
| Store | 88 | 97.8 |
| In the house/ground | 2 | 2.2 |

### 3.4. Mean Aflatoxin Levels

Table 4 is a summary of the mean aflatoxin levels in maize collected from farmers in the different field schools in the study area. Except for Kisob Katanin, there was a significant difference in the mean aflatoxin levels between the two maize planting seasons in all other study sites. Table 4.

**Table 4.** Mean aflatoxin levels (ppb) in maize sampled from different farmer field schools.

| Division | Farmer Field School | N | Range for Season 1 | Range for Season 2 | Mean for Season 1 | Mean for Season 2 |
|---|---|---|---|---|---|---|
| **Kaptumo** | Keteba | 20 | 1.4–3.6 | 2.8–3.9 | 2.3 ± 0.7 | 3.4 ± 0.4 * |
| | Mwangaza | 16 | 0.8–2.6 | 0.0–1.1 | 1.5 ± 0.5 | 0.3 ± 0.5 * |
| **Kilibwoni** | Kisob Katanin | 12 | 1.0–4.4 | 0.0–3.7 | 3.0 ± 1.1 | 2.5 ± 1.0 ns |
| | Toleltany | 9 | 1.0–4.0 | 0.0–3.7 | 2.6 ± 1.4 | 0.1 ± 3.5 * |
| **Kipkaren** | Sobetab Gaa | 15 | 0.0–2.9 | 0.0–0.9 | 1.3 ± 0.6 | 0.2 ± 0.4 * |
| | Toret Gaa | 10 | 0.0–1.3 | 0.0–0.8 | 0.8 ± 0.5 | 0.1 ± 0.3 * |
| | | **82** | **0.0–4.4** | **0.0–3.9** | **1.9 ± 1.1** | **1.3 ± 1.5 *** |

N: number of farmers, *: significantly different, ns: not significantly different; there was a significant difference in mean aflatoxin levels in maize between season 1 and season 2 in all the farmer field schools except for Keteba in the Kaptumo division.

## 4. Discussion

### 4.1. Socio-Demographics of Maize Farmers Interviewed in the Study Area

The statistics behind the practice of agriculture in Kenya are damning. Up to 25% of Kenya's gross domestic product is tied to agriculture, with the sector employing a remarkable 75% of the national labor force [26]. Maize is the staple food for most households in Kenya, and is mostly produced by small scale farmers [27]. It is also an important livestock feed, both as a silage and crop residue [27]. Grains are also used industrially for starch and oil extracts [27].

Our observations that there were more female farmers than male ones in Nandi County bodes well with the narrative perpetuated by Diiro and colleagues, who reported that there was a positive correlation between maize productivity in Western Kenya and women's empowerment in agriculture [28]. Our observation that most farmers had an alternative source of income other than maize farming may have something to do with the problems that have been facing maize farmers in Kenya for the last two years. These include drought, armyworm infestation, and lack of market access [29]. It could be suggested that farmers in the study area may have begun diversifying their crops, given the attendant problems associated with growing maize in the region for commercial gain.

### 4.2. Land Preparation, Weeding, Correct Spacing, Cleaning Stores, Sorting Maize after Shelling, Harvesting, and Moisture Content of Harvested Maize Seeds

Generally, practices such as weeding, checking the condition of maize, and sorting the maize after shelling were well received among the farmers interviewed.

All farmers did land preparation. According to the training manual, land preparation removes crop residues such as old stems that may be lying on the farm and that may serve as potential substrates for the growth of fungi responsible for aflatoxin contamination [30]. Moreover, it is noteworthy that all farmers interviewed made use of Kenya seed 6213, which is recommended for this agro-ecological zone. The advantage of this seed is that it produces better yields per unit area than other seed varieties, has a full husk cover which is important as rotting is minimized, and is moisture and disease-resistant.

All farmers interviewed practiced weeding on their farms. According to the training manual, weeding is important, as it removes substrates that facilitate the growth of mycotoxin-producing fungi. Moreover, weeding makes it so that aflatoxin-producing fungi are buried and scattered, thereby greatly reducing the population mass of the fungi and limiting its growth. Moreover, Atehnkeng and colleagues recommend that it is important to undertake timely control of weeds to avoid them competing with the crop. They advise weed control via the use of a hoe, bull, tractor or herbicide [31].

The training manual advocated that the correct spacing of seedlings should be 75 × 25 cm. All farmers took well to this aspect of the training. Overcrowding should be avoided, and optimum populations should be established to avoid creating drought stress for the crop [31].

All farmers interviewed cleaned their stores before storing their harvest. According to the code of practice for the prevention and reduction of aflatoxin in tree nuts, the cleaning of stores is important, as stores that are not clean and dry may facilitate fungal growth during storage [31].

All farmers interviewed reported that they sorted the bad maize from the good after shelling. According to the code of practice for the prevention and reduction of aflatoxin contamination in tree nuts [31], the sorting of maize is important, since aflatoxin-producing fungi have a better chance of contaminating damaged seeds than good seeds. Not sorting seeds is likely to facilitate the contamination of the stored crop with aflatoxin.

All farmers interviewed used the manual method of harvesting, rather than the mechanical method. According to the training manual, mechanical damage to the grain during harvesting exposes the seed to contamination due to the breakage of the seeds. This is particularly true when poorly calibrated threshers are used [32].

### 4.3. Early Planting, Crop Rotation, Use of Wooden Pallets, Stooking

About 90% of farmers interviewed reported that they did early planting. According to the training manual, early planting is recommended, as crops have been reported to be less affected by pests, toxins, and disease when planted early. In the case of maize, late planting exposes the crop to diseases such as maize lethal necrosis disease (MNLD) and maize stalk borer. Atehnkeng and colleagues recommend that the planting of crops should be done at the right time to enable crop disease escape and have enough rain for growth and maturity towards the end of the season [32].

About 92% of the farmers interviewed reported having practiced crop rotation. Crop rotation is important in reducing mycotoxin contamination in grains [32]. Also, crop rotation has been reported to reduce aflatoxin prevalence in crops by breaking the cycles and build-up of toxin-producing microorganisms [33]. This notwithstanding, it is quite disconcerting to observe that some farmers did not carry out this important measure. We posit that more should be done to further educate farmers in this region on the importance of this practice. Moreover, according to the training manual, planting the same commodity on a farm for 2–3 consecutive years may make crops susceptible to mycotoxin-producing fungi. It is also recommended that grains, particularly maize and wheat, should not be used in crop rotation with each other [32].

The use of wooden pallets to store maize was common. According to the code of practice for the prevention and reduction of aflatoxin contamination in peanuts [34], grain storage bags should be clean and dry and stacked on pallets, or a water-impermeable layer should be incorporated between the bags and the floor [34].

About 94% of all farmers interviewed reported that the size of their stook was small. According to the training manual, the size of the stook needs to be small to facilitate free air circulation. In West Africa, the practice of stooking is also referred to as heaping [32]. It is advised that not only should the heaps/stooks remain erect in the form of a cone, but also, the heaps should not be too big or fall on the ground, as they are likely to accumulate moisture at their center [32]. Moreover, it is advised that maize cobs that have been lying on the ground for long or those with signs of animal/insect damage should not be heaped/stooked [32].

About 88% of the farmers interviewed reported that the harvested maize was moderately dry. Atehnkeng recommends that maize should be dried to less than 13% moisture content without delay. However, since we did not measure the moisture content of the seeds harvested, it is difficult to know exactly how much moisture was in the seeds.

### 4.4. Rain during Harvesting, Storage of Maize, Fertilizer Use, Liver Cancer Related Mortality, Insecticide Use, Positioning of Stores, Feeding Rotten/Damaged Seeds to Animals

Reports of rain during harvesting from about 48% of farmers should be a cause of concern. A previous report on mortality associated with acute aflatoxicosis reported that the toxic contamination of maize may have taken place during a period when the maize harvesting season was blighted by heavy rains [18].

About 50% of the farmers interviewed reported storing maize on the ground after harvesting rather than on a mat or rack. This may have potentially exposed the crop to the soil which may have been carrying the *Aspergillus* fungi, and may have facilitated aflatoxin contamination.

According to Atehnkeng and others, crops grown under stress are more susceptible to infestation by the aflatoxin-producing fungi that cause contamination [31]. According to the same author, applying fertilizer and other key inputs reduces crop stress [31]. In our study, about 44% of the farmers interviewed reported using the 75 kg bag of fertilizer per acre recommended by the training manual, while about 52% reported using less than the recommended amount per acre. Another 3% reported using the 100 kg bag of fertilizer. Further research on the effect of the amount of fertilizer used per acre on aflatoxin levels in maize is warranted.

About 20% of farmers reported that they knew a relative who had died of liver cancer. Though associations have been drawn between aflatoxin exposure and the risk of liver cancer [34],

it is difficult to assert whether the person(s) the farmers were referring to had liver cancer as a result of feeding on aflatoxin-contaminated maize, or whether it developed as a result of other factors. Further work is needed to shed more light on this question.

About 28% of farmers did not apply insecticide to the harvested crop after shelling. Atehnkheng [31] advises that it is important to control insects, particularly the stem borer, as insects may damage the crop, leading to an invasion by aflatoxin-causing fungi. Moreover, Nesci and colleagues [35,36] have reported that insects may act as vectors for aflatoxigenic fungi, and not using insecticide may open up an avenue for aflatoxin contamination of harvested maize.

Only 24% of the farmers interviewed reported that they did not have stores that were raised above the ground. Failure to raise the stores above the ground may lead to increased exposure of the harvested crop to soil containing *Aspergillus* fungi, and could also improve the access of rodents to the harvested crop [37]. This access may lead to breakage of the maize seeds, which could make it easier for aflatoxin-producing fungi to contaminate the maize [37].

About 14% of the farmers interviewed reported that they fed damaged and rotten maize seeds to animals. *Aspergillus* infection has been shown to occur in broken and damaged kernels [2]. Thus, feeding animals with such seeds may expose them to aflatoxin poisoning. This observation implies that despite acknowledging that they were aware of aflatoxins and the damage it causes to humans and animals, it was lost on farmers that feeding rotten maize to animals may have been unnecessarily exposing them to harm. It is important to note that the food and drug administration has developed guidelines for acceptable aflatoxin levels in maize according to its intended use. Based on these guidelines, levels of <20 ppb, <20 ppb, <100 ppb, <200 ppb, and <300 ppb are considered appropriate for young animals, dairy cattle, breeding meat cattle/swine/mature poultry, finishing swine, and finishing cattle, respectively [34].

### 4.5. Confidence Intervals on the Response of Farmers in Nandi County

Questions on feeding damaged/rotten seeds to animals and harvesting during the rain had the lowest confidence intervals. To maintain or even lower the observed aflatoxin levels in maize grown in Nandi County, future training on GAP may need to focus on improving these aspects of GAP training. On the other hand, questions on crop rotation, early planting, and raising the stores above the ground had the highest confidence intervals. This suggests that these aspects of GAP were well understood by farmers in the study area.

### 4.6. Mean Aflatoxin Levels

In the first season, about three samples from an equivalent number of farmers had an aflatoxin level of 0, while in season 2, up to 55 samples from an equivalent number of farmers had an aflatoxin level of 0. This may suggest that the training on good agricultural practice seems to be paying off. It may be conjectured that if the farmers were to continue applying what they learnt from the training, more samples would be likely to record similar aflatoxin levels. Furthermore, the action level for aflatoxin-contaminated grains set by the Kenya Bureau of Standards is 10 ppb [38]. It is interesting to note that the levels of aflatoxin in the maize sampled from the farmers were all well below acceptable levels. This appears to be further evidence that the training was well-received among the farmers.

Farmers from the farmer field school in Keteba (Kaptumo division) reported that it rained for six months while the crop of maize was in the field in Season 2. This may explain why this was the only farmer field school that had mean aflatoxin levels that were higher in the second season than the first. A previous study reported that aflatoxin levels in maize were significantly associated with agro-ecological zones, and that humidity played a big part in the overall aflatoxin levels [38]. This may be the reason behind the observation of an increase rather than a decrease in aflatoxin levels of maize sampled from this area. Moreover, in some divisions, progress in the mitigation of aflatoxin risks could be observed, but not in all. It could be argued that practices such as crop rotation, early planting, and raising the stores above the ground may be responsible for this observation.

Our findings on the mean aflatoxin levels were comparable to those of Kang'ethe and coworkers, who reported baseline values of 0.997 ppb [15]. Moreover, the mean aflatoxin levels we have reported for both seasons were lower than the mean aflatoxin levels of 2.3 ppb reported by Mutiga and others in maize sampled from storage sheds and mills in Nyanza, Rift Valley, and Western Kenya [39]. This may also be further evidence that training farmers on good agricultural practice may indeed be having positive effects of lowering the aflatoxin levels in maize grown in the region.

## 5. Conclusions

Our findings suggest that some components of the training were better understood and more widely practiced by the farmers in Nandi County than others. Aflatoxin levels in maize evaluated before and after the training appear to suggest that training farmers on good agricultural practice may be an important tool in minimizing aflatoxin exposure in maize. Further engagement with farmers is needed, especially concerning the practices that were not very well understood and widely practiced. Continued monitoring of aflatoxin levels in maize in the region should be encouraged. We also recommend that a similar strategy be adopted in other maize growing regions in the country. Even though training farmers to adopt good agricultural practices was observed to be efficient in combatting the aflatoxin problem in maize in Nandi County, it can be anticipated that the most efficient mitigation measures will vary in other agro-ecological zones. Therefore, local applicability should be considered when evaluating the specific measures for designing an optimized regional aflatoxin contamination management strategy.

**Author Contributions:** Conceptualization: H.K., V.J., L.W.K., J.M.M., and G.N.M. Methodology: H.K., V.J., G.N.M., and L.W.K. Validation: L.W.K., M.O.O., J.M.M., and G.N.M. Formal analysis: M.O.O. and G.N.M. Investigation: All authors. Resources: H.K., V.J., P.A.A., and M.O.O. Data curation: L.W.K., G.M.M., and J.M.M. Writing—original draft preparation, M.O.O. and G.N.M.; writing—review, and editing: L.W.K., J.M.M., P.A.A., H.K., and V.J. Visualization: M.O.O. and G.N.M. Supervision: L.W.K., J.M.M., H.K. and V.J. Project administration: L.W.K., J.M.M., V.J., and H.K. Funding acquisition: H.K., V.J., and L.W.K. All authors have read and agreed to the published version of the manuscript.

**Funding:** This work was carried out under the financial support of the Ministry of Foreign Affairs, Government of Finland grant number 24821134.

**Acknowledgments:** The authors would like to acknowledge Joshua Orungo Onono of the Department of Public Health, Pharmacology and Toxicology of the University of Nairobi who assisted in data analysis. Special gratitude to Duke Omayio Gekonge for assistance in some aspects of the data analysis.

**Conflicts of Interest:** The authors declare no conflict of interest. The authors would wish to state that the funders had no role in the design of the study; in the collection, analyses, or interpretation of data; in the writing of the manuscript, or in the decision to publish the results.

## Appendix A. Questionnaire Used in Interviewing Farmers in the Study Area

*Appendix A.1. Section A: Sociodemographics*

- Gender: Male/Female?
- Number of years in farming: 1-5 years, 6-10 years, 11-15 years, >15 years?
- Time of the year when the maize was planted: January, March, April?
- Acreage of the farm: <1 acre, 1-5 acres, 5.5-20 acres
- Alternative sources of income: Yes/No
- Reasons for farming: Subsistence/commercial?

*Appendix A.2. Section B: Evaluation of the Knowledge of Farmers*

- Was top dressing done?
- Was stooking done after harvesting?
- Were stores raised above the ground?

- Were pallets available to place the maize in the stores?
- How was the maize shelled? Tractor/hand Sheller?
- What kind of storage bag was used for the maize?
- Have you heard about aflatoxins?
- Do you think aflatoxins are harmful to human beings and animals?
- Do you know any relative who has died of liver cancer?

*Appendix A.3. Section C: Attitudes*

- Was correct spacing done?
- Were fertilizer and manure used
- How much fertilizer was used per acre?
- How dry was the maize at harvesting?
- Was harvested maize placed on a mat/rack or the ground?
- Was the maize stored after harvesting or did it go to the ground of the house?

*Appendix A.4. Section D: Practices*

- Was early planting done?
- Was land preparation done?
- Was weeding done?
- Was the store cleaned before use?
- Was there any attempt to dry the maize after harvesting?
- Was sorting/drying done after shelling?
- Was insecticide used/applied after shelling?
- Was crop rotation practiced?
- Were damaged seeds fed to the animals?
- Was clean transport used to transport the maize?

## Appendix B. Supplementary Data

*Appendix B.1. I. Kaptumo Division*

**Table A1.** Paired Samples Statistics.

| | | Mean | N | Std. Deviation | Std. Error Mean |
|---|---|---|---|---|---|
| **Pair 1** | SEASON 1 | 2.3 | 20 | 0.7 | 0.2 |
| | SEASON 2 | 3.3 | 20 | 0.4 | 0.1 |

*a.* *Keteba*

**Table A2.** Samples Test.

| | | Paired Differences | | | | | t | df | Sig. (2-tailed) |
|---|---|---|---|---|---|---|---|---|---|
| | | Mean | Std. Deviation | Std. Error Mean | 95% Confidence Interval of the Difference | | | | |
| | | | | | Lower | Upper | | | |
| **Pair 1** | SEASON 1–SEASON 2 | −1.1 | 0.8 | 0.2 | −1.4 | −0.7 | −5.9 | 19 | *<0.05* |

*b.    Mwangaza*

**Table A3.** Samples Statistics.

|  |  | Mean | N | Std. Deviation | Std. Error Mean |
|---|---|---|---|---|---|
| **Pair 1** | SEASON 1 | 1.5 | 16 | 0.4 | 0.1 |
|  | SEASON 2 | 0.3 | 16 | 0.5 | 0.1 |

**Table A4.** Samples Test.

|  |  | Paired Differences | | | | | | | |
|---|---|---|---|---|---|---|---|---|---|
|  |  | Mean | Std. Deviation | Std. Error Mean | 95% Confidence Interval of the Difference | | T | df | Sig. (2-tailed) |
|  |  |  |  |  | Lower | Upper |  |  |  |
| **Pair 1** | SEASON 1–SEASON 2 | 1.2 | 0.8 | 0.2 | 0.8 | 1.6 | 6.1 | 15 | *<0.05* |

**Table A5.** Sample Correlations (Kaptumo division).

| Study Area |  | N | Correlation | Significance |
|---|---|---|---|---|
| Keteba | Pair 1 SEASON 1 & SEASON 2 | 20 | 0.1 | 0.8 |
| Mwangaza | Pair 1 SEASON 1 & SEASON 2 | 16 | −0.3 | 0.3 |

*Appendix B.2. II. Kilibwoni Division*

*a.    Kisob Katanin*

**Table A6.** Samples Statistics.

|  |  | Mean | N | Std. Deviation | Std. Error Mean |
|---|---|---|---|---|---|
| **Pair 1** | SEASON 1 | 3.0 | 12 | 1.1 | 0.3 |
|  | SEASON 2 | 2.5 | 12 | 1.0 | 0.3 |

**Table A7.** Samples Test.

|  |  | Paired Differences | | | | | | | |
|---|---|---|---|---|---|---|---|---|---|
|  |  | Mean | Std. Deviation | Std. Error Mean | 95% Confidence Interval of the Difference | | T | df | Sig. (2-tailed) |
|  |  |  |  |  | Lower | Upper |  |  |  |
| **Pair 1** | SEASON 1–SEASON 2 | 0.5 | 1.0 | 0.3 | −0.1 | 1.2 | 1.8 | 11 | *>0.05* |

*b.    Toleltany*

**Table A8.** Samples Statistics.

|  |  | Mean | N | Std. Deviation | Std. Error Mean |
|---|---|---|---|---|---|
| **Pair 1** | SEASON 1 | 2.6 | 9 | 1.4 | 0.5 |
|  | SEASON 2 | 0.1 | 9 | 0.4 | 0.1 |

**Table A9.** Sample Correlations (Kilibwoni division).

| Study Area | | N | Correlation | Significance |
|---|---|---|---|---|
| Kisob Katanin | Pair 1 SEASON 1 & SEASON 2 | 12 | 0.5 | 0.1 |
| Toleltany | Pair 1 SEASON 1 & SEASON 2 | 9 | −0.4 | 0.3 |

**Table A10.** Paired Samples Test.

| | | Paired Differences | | | | | | | |
|---|---|---|---|---|---|---|---|---|---|
| | | Mean | Std. Deviation | Std. Error Mean | 95% Confidence Interval of the Difference | | T | df | Sig. (2-tailed) |
| | | | | | Lower | Upper | | | |
| **Pair 1** | SEASON 1–SEASON 2 | 2.5 | 1.6 | 0.5 | 1.3 | 3.7 | 4.7 | 8 | *<0.05* |

*Appendix B.3. III. Kipkaren Division*

*a.    Sobetab Gaa*

**Table A11.** Paired Samples Statistics.

| | | Mean | N | Std. Deviation | Std. Error Mean |
|---|---|---|---|---|---|
| **Pair 1** | SEASON 1 | 1.3 | 15 | 0.6 | 0.2 |
| | SEASON 2 | 0.2 | 15 | 0.4 | 0.1 |

**Table A12.** Paired Samples Test.

| | | Paired Differences | | | | | | | |
|---|---|---|---|---|---|---|---|---|---|
| | | Mean | Std. Deviation | Std. Error Mean | 95% Confidence Interval of the Difference | | T | df | Sig. (2-tailed) |
| | | | | | Lower | Upper | | | |
| **Pair 1** | SEASON 1–SEASON 2 | 1.1 | 0.7 | 0.2 | 0.7 | 1.5 | 6.12 | 14 | *<0.05* |

*b.    Toret Gaa*

**Table A13.** Paired Samples Statistics.

| | | Mean | N | Std. Deviation | Std. Error Mean |
|---|---|---|---|---|---|
| **Pair 1** | SEASON 1 | 0.8 | 10 | 0.5 | 0.1 |
| | SEASON 2 | 0.1 | 10 | 0.3 | 0.1 |

**Table A14.** Sample Correlations (Kipkaren division).

| Study Area | | N | Correlation | Significance |
|---|---|---|---|---|
| Sobetab Gaa | Pair 1 SEASON 1 &SEASON 2 | 15 | 0.1 | 0.8 |
| Toret Gaa | Pair 1 SEASON 1&SEASON 2 | 10 | 0.3 | 0.5 |

**Table A15.** Paired Samples Test.

| | | Paired Differences | | | | | | | |
| | | Mean | Std. Deviation | Std. Error Mean | 95% Confidence Interval of the Difference | | T | df | Sig. (2-tailed) |
| | | | | | Lower | Upper | | | |
| **Pair 1** | SEASON 1–SEASON 2 | 0.8 | 0.5 | 0.1 | 0.4 | 1.1 | 5.1 | 9 | *<0.05* |

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
