# Peer review of "Effects of the Use of Good Agricultural Practices on Aflatoxin Levels in Maize Grown in Nandi County, Kenya"

_sci, doi:10.3390/sci2040085_

Round 1

Reviewer 1 Report

I appreciate the efforts of the authors. Anyway, they should provide further details. 

1) For example, how was the questionnaire developed? Please describe in detail each step of the questionnaire development.

2) Is the questionnaire validated?

3) Which are its psychometric properties?

4) The way of reporting statistcal analysis and results needs to be improved.

5) Figures and numbers should be rounded with the same level of precision. 

6) Authors could perform more statistical analysis. 

7) In which way were the participants recruited?

8) Better develop strengths and limitations of the research. 

Author Response

POINT BY POINT RESPONSES TO COMMENTS RAISED BY REVIEWERS Reviewer 1 Sent on 17 Jun 2019 by Nicola Luigi Bragazzi | Approved with revisions University of Genoa, Genoa, Italy I appreciate the efforts of the authors. Anyway, they should provide further details. 1) For example, how was the questionnaire developed? Please describe in detail each step of the questionnaire development. 2) Is the questionnaire validated? 3) Which are its psychometric properties? 4) The way of reporting statistical analysis and results needs to be improved. 5) Figures and numbers should be rounded with the same level of precision. 6) Authors could perform more statistical analysis. 7) In which way were the participants recruited? 8) Better develop strengths and limitations of the research. 1. Thank you. We have provided the questionnaire that was used in interviewing patients at the appendix section. Page 11, 12. Line 384-418. 2. The questionnaire was not validated, as it was used for gathering basic information for further research purposes. 3. The questionnaire was used to collect data based on the effect of agricultural measures on the aflatoxin levels in maize. Personal data collected was used only to elucidate the gender and age distribution of farmers, the correlation between the participating farmers and levels of aflatoxin was not examined in this study. 4. Thank you for this observation. We have summarized most of the data presented in figures into a single table. Table 2. Line 188 5.Thank you. We have rectified this in all sections of the manuscript. Tables 1-4. Line 179, line 188, line 193, line 199. 6. We have made a recap table that now summarizes all the data presented sporadically in sections 3.2-3.6. The table shows the specific agricultural measures (early planting, land preparation, correct spacing, etc.) taken, the number of farmers giving a reply, confidence interval of replies and - based on that – percentage of farmers who took the said measures. Table 2. Line 188 7. We have already described this in section 2.2. Study Design. 8. Thank you for the observation. We have added a statement in the conclusion section that addresses the strength and limitations of the research. i.e. ‘even though training farmers to adopt good agricultural practices was observed to be efficient in combatting aflatoxin problem in maize in Nandi County, it can be anticipated that the most efficient measures to mitigate the aflatoxin problem vary in various agro-ecological zones. Therefore, local applicability should be considered when evaluating the specific measures for designing an optimized regional aflatoxin contamination management strategy.’ Line 357-368.

Reviewer 2 Report

Dear authors,

your paper is fine and I like your relation between farmers' behaviour and aflatoxin levels. I would recommend to make a little bit more statistics, especially to find out which of the parameters explain the variability in aflatoxin levels at the best. Small comments are in the attched file.

Best regards,

Thomas

Author Response

Dear authors, 1. Your paper is fine and I like your relation between farmers' behavior and aflatoxin levels. I would recommend making a little bit more statistics, especially to find out which of the parameters explain the variability in aflatoxin levels at the best. Small comments are in the attached file. 2. Which dimension does this level have, e.g. mg/kg maize? 3. Please write species, not in italics 4. Please write in italics 5. Please give a short notice whether this substance is from which fungus species 6. What is this critical limit? 7. Does more than 100% make sense? 8. Please explain the level, is there a relation to mg/kg maize? ppb? 9. Please use n. s. for "not significant (** would be highly significant!) 10. One comment: is it possible to compare aflatoxin levels with farmer practice (crop rotation, harvest during rain, kind of storage) and to test the differences between the two groups ("yes" and "no"). Has one of the parameters from the questionnaire the most prominent effect on the aflatoxin level? 11. You have results on possible correlations but you do not report them, why? Thank you for this observation. We did not report on the correlations as all of them were non-significant. 1. Thank you very much for your kind words. We have made a recap table that now summarizes all the data presented sporadically in sections 3.2-3.6. The table shows the specific agricultural measures (early planting, land preparation, correct spacing, etc.) taken, the number of farmers giving a reply, confidence interval of replies and - based on that – percentage of farmers who took the said measures. Table 2. Line 188. 2. Thank you for the observation. The dimension was ppb. Page 1: line 35 3. Thank you. We have made the change. Page 2; line 43 4. Thank you for the observation. We have made the change. Page 2 line 45 5. Thank you for the suggestion. We have made the change. Page 2 line 56 i.e. fumonisins (a group of mycotoxins derived from Fusarium). 6. Thank you for this keen observation. The critical limit is 20 ppb. We have made the change. Page 2. Line 71. 7. Thank you for this very critical observation. We have re-calculated all the confidence intervals and have rectified all the discrepancies and summarized the confidence intervals calculated into a table (Table 2. Line 188) 8. Thank you for this observation. We have clarified that the levels were expressed in parts per billion. Conventionally, 1ppb is equivalent to 1mg/kg 9. Thank you. We have made the change to reflect your recommendation. Page 7. Line 199 10. This is a very good suggestion. However, it was beyond the scope of our study. We were not able to individualize the aflatoxin levels we have reported so that a distinction could be made between farmers who responded ‘yes/no’ to certain practices and the corresponding aflatoxin levels. We identify the limitation and have tried to explain it in the discussion section. Page 10. Line 347-349 11. Thank you for this observation. We did not report on the correlations as all of them were non-significant.

Round 2

Reviewer 2 Report

Dear author(s),

only a small number of writing errors have been found (which I didi not detect in the first version).

Best regards,

Thomas